# Orthotic management of instability of the knee related to neuromuscular and central nervous system disorders: qualitative interview study of patient perspectives

Dorothy McCaughan,[1] Alison Booth [iD],[2] Cath Jackson,[1] Simon Lalor,[3,4] Gita Ramdharry,[5,6] Rory J O'Connor,[7] Margaret Phillips,[8] Roy Bowers,[9] Catriona McDaid [iD] [1]

For numbered affiliations see end of article.

**Correspondence to**
Mrs Dorothy McCaughan;

## ABSTRACT

**Objectives** Adults with knee instability related to neuromuscular disorders or central nervous conditions often experience mobility problems and rely on orthoses to improve function and mobility. Patient views of device effectiveness and acceptability are underexplored. Our study aimed to elicit device users' perspectives regarding fitting, acceptability, effectiveness and use of orthoses, and identify important treatment outcomes.

**Design** Qualitative descriptive study using in-depth semistructured interviews. Interview transcriptions were coded and thematically analysed, using 'Framework'.

**Setting and participants** A purposive sample of 24 adult users of orthotic devices. Nineteen patients were recruited across three National Health Service sites, and five people through charities/patient support groups in England. Half of the participants had been diagnosed with poliomyelitis, and the remainder with multiple sclerosis, Charcot-Marie-Tooth disease, spinal injury or spina bifida, and stroke. The median age of participants was 64.5 years (range 36–80 years).

**Results** Patients' medical condition impacted significantly on daily life. Participants relied on orthotic devices to enable engagement in daily activities. Patient goals for mobility were linked to individual circumstances. Desired treatment outcomes included reduction in pain, trips and falls, with improved balance and stability. Effectiveness, reliability, comfort and durability were the most valued features of orthoses and associated with reported use. Obtaining suitable footwear alongside orthotic devices was a significant concern. Time pressures during device fitting were viewed negatively.

**Conclusions** Orthotic devices for knee instability play a crucial role in promoting, maintaining and enhancing physical and psychological health and well-being, enabling patients to work, engage in family life and enjoy social activities. Future research should consider how best to measure the impact of orthotic devices on patient quality of life and daily functioning outside the clinic setting, as well as device use and any adverse effects.

**Trial registration number** This qualitative study was retrospectively registered as Current Controlled Trials ISRCTN65240228.

### Strengths and limitations of this study

► This is the first study specifically designed to explore the views and experiences of patients regarding effectiveness and acceptability of orthotic devices, and to identify outcomes that are important for people who have been fitted with an orthotic device for knee instability, across a broad range of conditions.
► Our study captured rich data concerning issues identified as significant by participants, providing important new insights.
► A full range of neuromuscular and central nervous system conditions (eg, myasthenia gravis and muscular dystrophy) were not represented in the sample.
► Nonetheless, the study included people with regular and sustained contact with orthotic services in connection with knee instability.

## INTRODUCTION

Adults with knee instability related to neuromuscular disorders (NMD) or central nervous system (CNS) conditions may experience pain, loss of balance, falls and mobility problems due to primary impairments of muscle weakness and/or sensory impairment. Muscle weakness can lead to instability of the joints which can be particularly problematic in the lower limbs during weight-bearing tasks; if the muscles cannot generate sufficient force to resist gravity, then the lower limb joints can be unstable or collapse. In addition, if sensory feedback is also impaired, people may be unaware of the position of their joints or if they have moved, so increasing the risk of instability.

Prescription of devices to stabilise the knee will depend on which muscles are weak that act on the knee joint. Knee–ankle–foot orthoses (KAFOs) are more often prescribed

when proximal lower limb weakness contributes to knee instability, for example, weakness of the quadriceps. The KAFO will stabilise the position of the femur on the tibia and the tibia over the foot, thus stabilising the knee. If lower forces are required and the ankle is stable, knee braces can be used to stabilise the femur on the tibia to support the knee. In the case of focal distal weakness, for example, the plantar flexal muscles, ankle–foot orthoses (AFO) can stabilise the ankle during stance, so maintaining the position of the tibia against gravity and preventing a collapse of the knee into excessive flexion or hyperextension.

Orthotic devices can offer support, align or correct deformities, or improve function.[1 2] A KAFO can be prescribed when other forms of bracing, such as an AFO or knee orthosis, are insufficient to adequately control knee instability, due to weakness or muscle imbalance. However, usage can be low.[3–5] Accurate figures relating to the number of people treated with orthotic devices are lacking, reflecting the challenges associated with obtaining data on orthotic services in England, partly due to the complexity of pathways of care.[6 7]

Literature concerning the views and experiences of users of lower limb orthotic devices for knee instability is scant.[8] A review of 10 studies[9] examined the patient use of lower limb orthotic devices (eg, knee brace and AFOs) and orthopaedic shoes, but none of the included studies related to use of KAFOs. The authors reported a wide range of rates of non-use, mainly due to pain, discomfort and non-cosmesis. A recent survey of orthoses users in Belgium[2] similarly reports varying levels of satisfaction with devices. While inclusion of 'open' questions in the survey elicited 'free-text' responses from participants, the underlying reasons for device satisfaction and use were not explored in depth, as this requires the use of appropriate qualitative methods. Results from focus group studies have appeared recently, but do not elucidate KAFO-users' perspectives. Swinnen et al[10] conducted 4 focus groups with 20 patients diagnosed with multiple sclerosis that revealed that stigmatisation, difficulties in putting on their lower limb orthosis (AFO), and aesthetic aspects were implicated in non-use. A focus group with eight AFO users living in the Netherlands to gain insight into importance of device-related activities to AFO users found that participants ranked walking as the most important of 11 different activities.[11] However, this latter small-scale study did not include any KAFO users, whose priorities for treatment outcomes may differ.

Understanding of the treatment outcome priorities of people with knee instability related to neuromuscular and CNS conditions is limited, and the range of different outcome measures used to evaluate patient outcomes following device fittings[12 13] makes it difficult to draw conclusions about effectiveness as data cannot be pooled for meta-analysis. Our study aimed to: (1) explore patient perceptions regarding fitting/acquisition, acceptability, effectiveness and usage of orthoses, and (2) identify the outcomes that are important for people who have been fitted with an orthotic device for knee instability, across a broad range of conditions. The qualitative study reported here comprises one strand of a larger study.[8] The report here focuses on treatment goals and outcomes of importance to patients, and device acceptability, contextualised within individuals' experiences of their specific medical condition. Further details of patients' views and experiences of orthotic service delivery can be found in the report of the overall study, which includes results from a systematic literature review, a survey of healthcare professionals and a costing analysis of KAFOs.[8]

## METHODS
### Study design
Semistructured individual interviews were undertaken (during 2014–2015) with adult users of orthotic devices. Qualitative approaches are well suited to the investigation of phenomena about which little is known, while semistructured interviews offer flexibility in data collection and generate rich narratives, allowing the researcher to analyse how participants make sense of the topic under investigation.[14] We followed the Consolidated Criteria for Reporting Qualitative Studies guidelines for reporting qualitative research.[15]

### Patient and public involvement
The study was designed in collaboration with a patient advisor, herself a user of an orthotic device, who contributed to the design and piloting of the interview topic guide (online supplementary appendix 1).

### Sampling
Inclusion criteria for the study were adults (≥16 years for National Health Service (NHS) participants; >18 years of age for non-NHS participants) with an NMD who have impaired walking ability primarily due to instability of the knee. NMD included conditions that primarily affect the peripheral nerve, muscle and neuromuscular junction, for example, motor neurone disease, muscular dystrophy, myasthenia gravis, spinal muscular atrophy, Charcot-Marie-Tooth (CMT) disease, poliomyelitis, myopathies and inclusion body myositis. People with knee instability that was related to CNS conditions were also included, for example, spinal cord injury, spina bifida and stroke. Participants were people who were able to give informed consent.

Participants were purposively sampled[16] to reflect a range of conditions, ages, gender, length of time fitted with an orthosis and regions across England. Potential participants who met the study inclusion criteria were approached by clinicians who were known to them, in three orthotic service/medical rehabilitation NHS outpatient settings: one located in the north, one in the south and one in the middle of England. Patients were informed about the study by the clinicians; those who expressed an interest were given written information about the study, including the researchers' contact details if they wished

to learn more and/or discuss taking part in a qualitative interview. Additionally, study information was provided to the chairpersons/lead representatives of the British Polio Fellowship, Charcot-Marie-Tooth UK, the FSH (facioscapulohumeral muscular dystrophy) Support Group UK and the Muscular Dystrophy Campaign, with a request to forward this to their members, who were invited to contact the qualitative researcher directly. Twenty-four people participated, nineteen recruited through NHS clinics and five through charities/patient support groups.

Participants were adults diagnosed with a neuromuscular or CNS condition who had been offered an orthotic device for knee instability of varying degrees of severity, that negatively affected their gait. Twelve had been diagnosed with poliomyelitis, and the remainder with multiple sclerosis, CMT disease, spinal injury, stroke and spina bifida. Participants' ages ranged from 36 to 80 years (median 64.5 years). Half worked part or full time while the remainder were retired; most (18) lived with their spouse and/or other family members. Further details relating to duration and severity of patients' conditions, and reported device use, can be found in online supplementary appendix 2: Participant Characteristics.

Written consent was obtained from all patients who took part in an interview.

## Data collection

A topic guide was developed, based on the aims of the research to explore patient perceptions of using an orthosis and goals for treatment; we sought to develop an instrument sufficiently structured to ensure consistency in information gathering, but flexible enough to allow participants to recount their individual experiences.[17] Input from the patient advisor helped guide relevance of content and comprehensiveness of the topic guide. Two experienced qualitative researchers (DM and CJ) undertook the interviews. Twenty-one interviews were conducted in participants' homes; three people requested a telephone interview as this was more convenient. In two cases, family members were present during the interview and their comments were included in data analysis. Interviews lasted approximately 1 hour and all were audio recorded. Data collection continued until no further pertinent information was forthcoming (after 24 interviews). Audio recordings were transcribed verbatim by an experienced transcriber, and a random sample of six transcripts checked for accuracy by DM and CJ.

## Data analysis

Data were analysed for thematic content using the 'Framework' method[18] (see figure 1). Framework is a flexible tool, not aligned with a particular epistemological, philosophical or theoretical stance, but adaptable to various qualitative approaches aiming to generate themes. Framework approach incorporates 'charting' or tabulation of data, which involves summarising and rearranging data according to thematic content, enabling the analyst to easily see the range of data across cases and

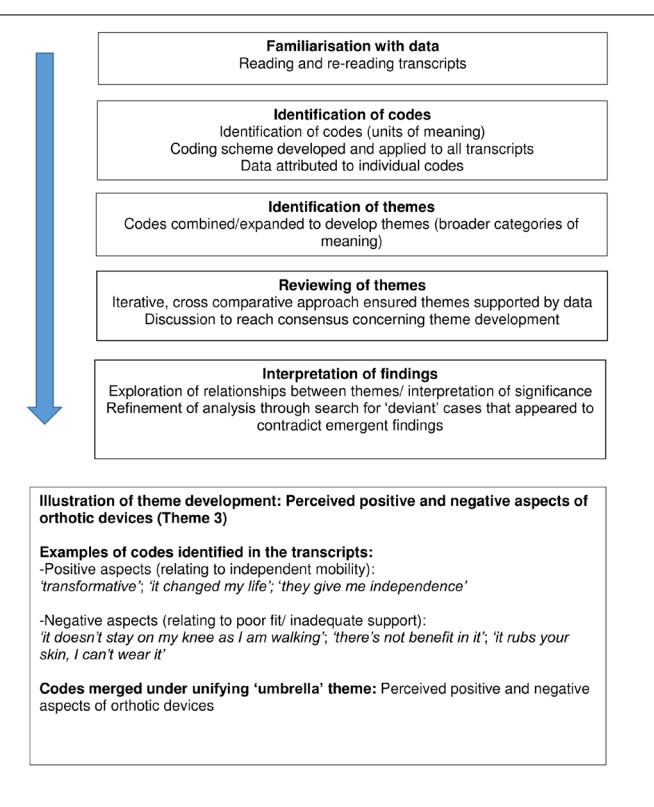

**Figure 1** Data analysis using framework approach.

under themes.[19] NVivo software package[20] facilitated data handling and retrieval and enabled comparison within and between cases. Analysis was systematic and iterative[19] and involved close reading and detailed coding of transcripts. Guided by the research questions, our approach to data analysis balanced both inductive and deductive orientations, as we sought to transfer the 'raw' data into a new and coherent description of the phenomena under scrutiny.[21] To promote analytical rigour, coding was cross-checked in four early transcripts by the two experienced qualitative researchers (DM, CJ) involved in the study. We sought to identify commonalities and differences in the data, with the aim of drawing descriptive and/or explanatory conclusions clustered around themes.[17] Differences in identification and development of themes were discussed until consensus was reached, and the thematic framework accordingly modified and expanded to accommodate all participant responses across the data set. Examination of negative or 'deviant' cases[22] helped to develop and refine the analysis.

## RESULTS

Five principal themes were identified: (1) impact of NMD and CNS conditions on walking and mobility; (2) fitting/acquisition and use/wearing of orthotic devices; (3) perceived positive and negative aspects of orthotic devices; (4) footwear; (5) desired treatment goals and outcomes. These themes are described below. Details

---

**Box 1    Symptoms and sequelae associated with neuromuscular disorders or central nervous system conditions**

**Reported symptoms**
► Pain (sometimes severe) in knee or ankle joints
► Hyperextension of knee joint
► Muscle weakness in the lower limb and/or of the muscles supporting joints
► Limbs of unequal length
► Feet 'frozen' in abnormal position
► Varying degrees of paralysis in lower limbs and feet
► Toes that 'curl inwards'
► Drop foot
► Fatigue

**Reported sequelae**
► Frequent falls, often preceded by feeling that leg is about to 'give way' under them
► Loss of sense of balance and stability
► Lop-sided gait
► 'Dragging' of feet
► Frequent trips, especially on uneven surfaces
► Difficulty standing for long or walking any but short distances
► 'Wear and tear' in unaffected or 'good' limb due to transfer of weight and effort in walking

---

relating to further themes/subthemes can be found in the main report.[8]

## Impact of NMD and CNS conditions

Participants' mobility was impaired by a range of symptoms and sequelae (see box 1) associated with their condition, including pain, muscle weakness, fatigue, loss of balance, unsteady gait and frequent trips and falls.

Limited mobility was associated with feelings of fearfulness and anxiety, resulting in diminished self-confidence and independence. People in employment were particularly anxious to gain or retain mobility as far as their condition allowed. Many participants described being proactive in trying to postpone or abate the impact of future deterioration on independent mobility, by keeping their weight under control, exercising regularly and undertaking exercise programmes recommended by physiotherapists. Reduced mobility was said to limit the pursuit of enjoyable activities such as gardening and walking, and could lead to social isolation. Retaining the ability to drive was viewed as important for both work and leisure purposes.

'The last time I was on the tube I got pushed...and I lack confidence and get scared, so if I *haven't got somebody with me, I won't use the tube.*' (Participant (P) 23)

*'I was in a very well paid job... and I saw how much I deteriorated in the last 5 years. ...I try to help myself...like with my weight... I have to work on that and my exercise ... I am active.'* (P22)

'Obviously I walk slower, so I am on my own quite a bit because I have got good friends who *will walk with*

*me, but a lot of people [won't] ... so it does actually make you an alone sort of person'* (P2)

'I've got an automatic car ... I've got a mobility car with a hoist in the back which can pick up *my scooter.'* (P18)

Unsurprisingly, obtaining the 'right' orthotic device(s) to help with current and future mobility emerged as a central concern of all study participants.

## Fitting/acquisition and use/wearing of orthotic devices
### Fitting/acquisition of KAFOs

Participants who used a KAFO (n=12) all had a diagnosis of poliomyelitis. These participants emphasised the importance of having a 'spare' KAFO for use during the period of being fitted for, and 'breaking in', a new device. Fitting was frequently described as a slow process, involving numerous appointments, often spaced at long intervals, and sometimes cancelled at short notice by the service provider. Attending appointments could mean time away from work, due to lack of 'out of hours' appointments, and some people mentioned difficulties with transportation. The fitting of a new device could be particularly challenging for those with complex condition-related problems; some individuals recalled attending multiple appointments stretching across a 2–3 year period before finally obtaining a device that fully met their needs and was comfortable.

*'If you're not getting the appointments, you're not getting the work done'* (P19)

'It was not a standard case, it made it much harder and I think for those first 2 or 3 years where it seemed I was going backwards and forwards to [clinic] and I felt that no real progress was being *made'* (P5)

Many people mentioned the importance of having sufficient time during fitting appointments. 'Double' appointments were said to allow more time for assessment and minor adjustments to devices that promoted optimal fit. Time pressures were perceived as constraining orthotist–patient communication regarding the new device and any adaptations to gait that might be needed. Problems might only come to light after the patient returned home and they could wait some considerable time for another appointment, unable to use the issued orthosis in the interim. Adequate time for full discussion during appointments was highlighted as particularly important to people receiving a KAFO for the first time, who might be struggling to accept and adapt to their orthosis.

'You don't really sort of pick up that sort of a problem when you're having a short *appointment...you're only there for 20 min, half an hour...you only think about the questions afterwards'* (P14)

A minority of participants reported receiving a device so ill fitting that it was unwearable; participant 15, for example, reported seeking a new KAFO through private

(non-NHS) suppliers due to the poor fit of their NHS device.

*'It wasn't a question of being uncomfortable, it didn't fit. It just did not fit and they took it away—I didn't even leave with it.'* (P8)

'It was too big, too high, so it was going right into my buttocks, it was too wide, so I wasn't getting *support at the knee and it was like it wasn't made for me.'* (P15)

Participants underscored the importance of being fitted for new footwear (discussed below) at the same time as being fitted for a new KAFO.

### Fitting/acquisition of AFO and knee braces

Four of the five AFO users were satisfied with the fit of their device. The fifth described her new AFOs as *'too painful to wear'* and causing damage to her skin, and she reverted to using her 'old' devices while waiting for the new ones to be adjusted.

*'I've just had a new pair … at the moment I've been bedding them in but something is not quite right with them … I've worn them for about a month but my foot has started to hurt again and I think the padding is not quite in the right place…'* (P21)

Two people (one diagnosed with a spinal injury, one with stroke) had been supplied with 'off-the-shelf' knee braces through physiotherapy services, but found them ill fitting and uncomfortable, so that they remained unused.

*'I just couldn't bear it. When I walked it just dug in, you can see the bruising…I would have preferred a bespoke one right from the start'* (P4)

### Use/wearing of orthotic devices

Device usage ranged along a continuum, from full-time use during the day, to little, or no use. Usage was closely linked by participants to capacity for independent mobility and therefore crucial to daily life. Many described their orthosis as integral to their social identity, enabling them to fulfil their desired roles as employee, spouse/partner, parent and friend.

'I need my surgical appliance to get on with my life …it needs to be effective and reliable …I want *to be in the real world. I love it …I'm in a working world and I just want to get on with it*' (P2)

Consequently, device malfunction was highly disruptive to 'normal' life.

'if it breaks, you're kind of thrown really …I came out of work, it was about half past six, and I got *across the road and the calliper just went, it just snapped*' (P6)

'As soon as this thing (KAFO) breaks down, I'm in silly street …that's not a place I want to go' (P19)

Other people reported using their device depending on the circumstances; for example, using their device outdoors, but not when at home. Intermittent use was

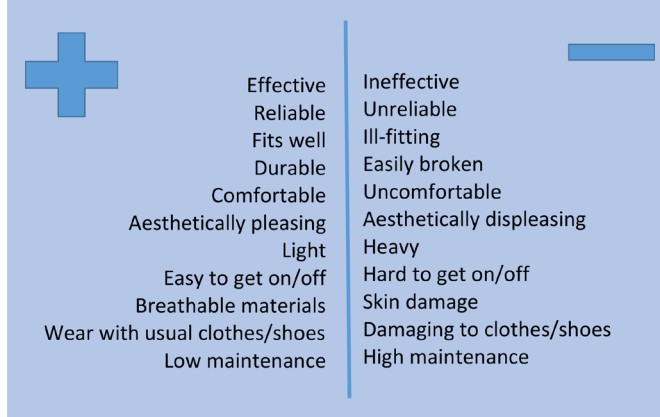

**Figure 2** Study participants' perceptions of positive and negative aspects of their orthoses.

also reported during *breaking in* of a new device, including alternating between wearing the new and old devices. Reluctance to make the transition from an old to a new device was widespread, due to '*teething troubles*' with the new device, and/or because of the need for gait adjustment, described by participant 18 as *'like having to learn to walk again'.*

Three of the 24 participants reported almost no use of their device as it did not offer adequate support for their knee or ankle. Additionally, a poor fitting device could result in skin damage and/or pain and discomfort, resulting in non-use. The most extreme (and therefore atypical) report of receipt of an ill-fitting orthosis came from participant 1 (diagnosed with poliomyelitis in childhood), who commented that she had never worn any of numerous orthoses provided for her, as they had been ineffective, ill fitting and uncomfortable to wear. During interview, participant 1 retrieved from elsewhere in the house the KAFO that she had most recently received through orthotic services, still in its plastic wrapper, and described by her as *'a complete waste of NHS money'*. Instead, she preferred to rely on a stick to help her mobilise, while recognising that this was not a good long-term solution, and she said she was finding it increasingly difficult to mobilise.

### Perceived positive and negative aspects of orthotic devices

Unsurprisingly, participants' perspectives of positive and negative aspects of their orthosis (see figure 2) synchronised with their reports of device use or non-use. Functionality and reliability were overwhelmingly considered more important than cosmesis, though appearance of the device was a significant feature for some of those interviewed (both male and female), who described the *'ideal'* device as light, sleek and discreet.

### Aspects of devices viewed positively

Effectiveness of the device to control pain and offer support for the knee or ankle, and/or to assist with lifting the foot, alongside reliability and durability, were the features viewed most positively. Reliability of the device

to withstand wear and tear was highly valued. The people most satisfied with their orthosis referred to the *'transformative'* effects on their lives of a well-fitting and well-functioning device.

> *'it's* [KAFO] *really transformed my life…from here to [place name] may not appear sort of a long distance but for me I don't think I could have done it beforehand, now I can which is a major sort of step forward …it's made me much more mobile than I ever was before really'* (P5)

> *'I was frightened to go out … It's just like, they [AFOs] just give me a new lease, you know, they give me my independence to go out on my own…with these splints I've got my independence. I can go out without my husband, I can go out with my friends.'* (P21)

Appearance, lightweight and 'user friendliness' (the ease with which the device could be donned and doffed, fastenings that are easy to manage and reliable, ease of use with preferred clothing and 'breathability of the materials used) were secondary concerns, though also appreciated.

> *'he showed me one at the time, which looked pretty cool…it just looked really, really good.'* (P8)

> *'It has carbon fibre, so it's lighter…it's not as wide, I can use most trousers'* (P3)

> *'modern day materials are absolutely fabulous…there is no weight in them and they are very strong'* (P22)

> *'It's speedy…you can put it on quickly'* (P6)

### Aspects of devices viewed negatively

Devices providing inadequate knee support, that were poor fitting (resulting in pain, discomfort or skin damage), and/or prone to breakage or malfunction, were described in strongly negative terms.

> *'this doesn't stay on my knee, as I am walking it falls right to the bottom…as I am walking it's slipping down…If it worked, it would be OK …there's no benefit in it…it just doesn't do nothing …I swear, I feel completely hopeless'* (P1)

> *'we went to [city] and literally we'd just landed at [name] airport and I'm walking to get my suitcase and I fell flat on my face …the thingy here, see there, it just went. Really is this screw holding me, I mean can't you think of something stronger than that!…the whole thing is very basic… the holiday was ruined…I couldn't relax…it was a short trip.'* (P23)

Breakage or device malfunction represented a crisis, temporarily depriving patients of their independence and interfering with plans and activities. Having a spare device in this situation was important to the majority of patients. They pointed out that even a seemingly simple repair, such as having leather straps replaced when they wear out, could result in difficulties and inconvenience. Other concerns related to discomfort, due to the device being heavy, bulky or cumbersome, devices considered aesthetically displeasing, difficult to put on or take off,

and those that caused damage to clothing and/or footwear. Dislike of hook-and-loop fastenings was common; they did not always hold the device securely, could 'snag' on tights and collect fabric fibres and fluff making them ineffective. Restrictions in the choice of clothing and/or footwear to accommodate devices could be irritating; three male participants reported wishing to appear smartly dressed on formal business or social occasions, but having to wear wide-legged trousers to accommodate their device.

> *'I have buckles, I hate Velcro'* (P16)

> 'it's difficult to get trousers that are really wide …so I buy women's jeans …from second-hand *shops, the charity shops …callipers damage them, I can't pay £50– £70'* (P16)

Serious skin damage in relation to wearing a KAFO or AFO was rarely reported, though marking or bruising of the skin in connection with newly acquired and/or ill-fitting devices was commonly mentioned, particularly where the knee is in close proximity to the joint on the KAFO. Skin rashes were mainly associated with plastic devices. Examples of 'home-grown' measures taken to prevent discomfort and minor skin damage included wearing tights, leggings or pyjamas under the device, adding extra padding at pressure points or, in one case, buying ankle straps that were made of softer leather than those supplied. Warm weather could give rise to skin irritation and several people mentioned looking forward to taking their device off at the end of the warm day to allow their skin to *'breathe'.*

> 'During the summer I quite often wear pyjama bottoms underneath the straps and leather of the *calliper'* (P9)

> *'One year…it was very hot… and I didn't have any tights on… it just rubs your skin, you can't wear it against the skin… I had blisters'* (P18)

> *'It's a bit like taking your shoes off … when I take the calliper off the skin can breathe a bit more'* (P3)

### Footwear

Being fitted for new shoes at the same time as for a new orthosis was considered important and positive and negative aspects of footwear were mentioned by participants (see figure 3). KAFO users diagnosed with poliomyelitis explained they might require each shoe of a pair to differ in width or height as their specific requirements can change over time, and require frequent reassessment.

> '*I had to insist and say, look none of your shoes is fitting me properly …I haven't had them reviewed since the 90s and I'm a lot older …every time I come, something is wrong.'* (P15)

Delays to the manufacture and delivery of new shoes were experienced as frustrating and disappointing. Shoes were said to require frequent repair as heels and soles can wear out quickly. Participants disliked having to rely

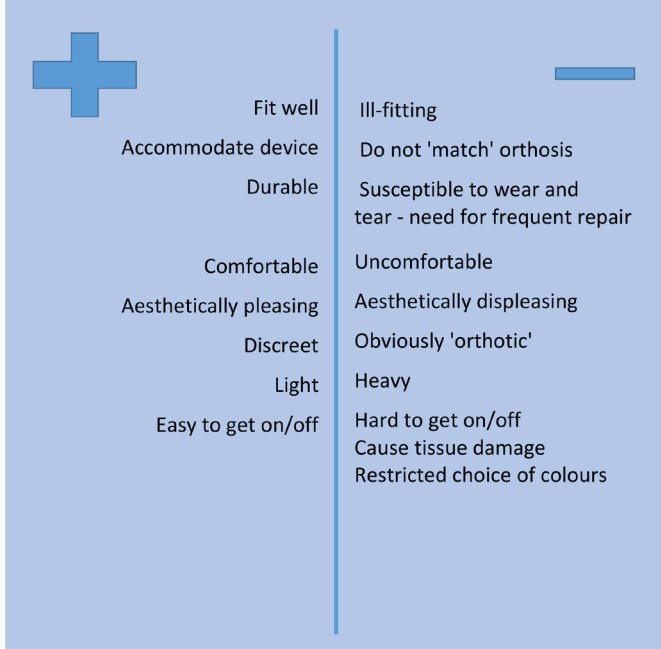

| Fit well | Ill-fitting |
| Accommodate device | Do not 'match' orthosis |
| Durable | Susceptible to wear and tear - need for frequent repair |
| Comfortable | Uncomfortable |
| Aesthetically pleasing | Aesthetically displeasing |
| Discreet | Obviously 'orthotic' |
| Light | Heavy |
| Easy to get on/off | Hard to get on/off |
| | Cause tissue damage |
| | Restricted choice of colours |

**Figure 3** Study participants' perceptions of positive and negative aspects of footwear.

on one pair of shoes for everyday use while repairs were being carried out on their other pair; many people said they usually owned only two pairs of orthotic shoes at any one time, regarding two pairs as their maximum entitlement. A good-fitting shoe that avoided damage to skin or deeper tissues was prioritised. Comfort was generally valued over appearance, although many people expressed a preference for footwear described as *'discreet'*. Heavy or *'clumpy'* orthotic shoes were said to interfere with ease of walking, and to draw attention to their condition. Several participants commented that the grip on the soles of shoes supplied to them was often inadequate for walking on uneven, muddy or icy surfaces, and suggested that a walking-shoe type of sole could improve confidence with walking outdoors.

*'there is no point in weeping over kitten heels because I am never going to wear those. But something reasonably presentable, that is not too intrusive, because you've got enough to be getting on in life with without having some people staring at things you might be wearing because obviously I am a bit conscious of the way I do walk … and people do stare a bit, so something that can be quite discreet is important.'* (P2)

Having a wide choice of colour, fastening (some people found hook and loop easier to use than laces), material (leather or suede) and style of shoes were appreciated, and most people thought that choice and quality of orthotic footwear have improved in recent years. Several people mentioned usually choosing black or brown shoes to '*go with anything*' and as suitable for work, while a few thought these were the only colours available to them. One person insisted on brightly coloured shoes to suit his personality.

*'Because of my personality, I like bright shoes, for example, I always wore bright green shoes … these are quite bright red really…it is part of your identity.'* (P5)

Participants using other types of orthoses (such as a knee brace) reported that they usually purchased shoes from local shops selling footwear, preferring particular brands of 'sturdy' shoes. Other people reported using orthotic devices such as insoles and inserts inside 'normal' shoes, either supplied through orthotic services or bought from commercial suppliers.

### Desired treatment goals and outcomes

Effective support for the affected (knee or ankle) joint with concomitant reduction of joint pain was the prime goal for most participants. Individuals wished to be as confidently mobile as their condition would allow, achieved through reduced anxiety that their joint might suddenly *'give way'*, and avoiding trips and falls. Preventing future deterioration of mobility (as far as feasible in relation to their specific condition) was a significant long-term goal. It was clear from participants' accounts that their desired treatment goals related to both physical and psychosocial aspects of their everyday lives. Cited benefits of independent and safe mobility included increased employment opportunities, ability to participate in day-to-day family activities and social events with friends and colleagues, as well as independent travel, and regular exercise, all regarded as vital to current and future physical and mental well-being.

*'I just do not want to become housebound, so that primarily is the objective of my orthotics.'* (P22)

*'I need my surgical appliance to get on with my life … it needs to be effective and reliable'* (P2)

*'It's just enabling me to keep going as long as possible which is really important to me.'* (P10)

Most study participants seemed to have accepted that they were unlikely to ever be able to walk very fast, or for very far, without some degree of fatigue, and they showed little interest in increasing the speed of their walking, or distance covered, as treatment goals per se. Instead, goals were expressed in terms of enablement of mobility, within the context and confines of individual circumstances, such as being able to work full time and undertake worldwide travel, walk a few hundred metres to nearby shops or being able to take just a few steps.

*'My walking is so limited. If I can stand and get something out of a cupboard and walk a few steps and get back to the wheelchair or whatever, that's what I can do, and that's what I need to do. For example in [name] mother's house I can't get the wheelchair in … because of the threshold, so I have to get out, walk a couple of steps over the threshold and then get back in the wheelchair and to be able to just walk those few steps to stand is an enormous advantage.'* (P16)

## DISCUSSION

### Principal findings

Participants described knee instability as compromising their ability to pursue desired daily activities and as having a negative impact on their social life. Effectiveness, reliability, comfort and durability were the most valued features of orthoses, and seemingly closely linked to their use, while cosmesis was perceived as of secondary importance. Reduction in pain, trips and falls, and improved balance and stability, promoting independent mobility were the most important outcomes to participants. The standard used by many participants to assess the effectiveness of their device was the extent to which it enabled them to engage in a broad range of activities viewed as important for physical and mental well-being. For some, even limited independent movement was regarded as a valued outcome. Comfort (fit and weight) of shoes, appearance and availability of choice regarding colour, materials, types of fastenings and style were all considered important by participants in our study, factors that seemed to be associated with likely wearing of prescribed shoes. Male participants were as equally concerned as females with cosmesis of devices and shoes. Participants highlighted various issues relating to the fitting/acquisition of devices, such as long waits between appointments, and time pressures during appointments, that resulted in reduced opportunities for contact and communication with clinicians. As a result, the period over which fitting took place might be prolonged, and patients were sometimes left feeling they did not receive the support and education they required to optimise adaptation to their device, especially where the device was a KAFO.

### Results in the context of other studies

As part of the same project, we also undertook a systematic review of the evidence on effectiveness of orthotic devices for the management of knee instability in adults with a neuromuscular or CNS disorder.[23] The review found that effectiveness studies focused on outcomes such as gait quality and energy consumption, assessed in the clinic setting. While important outcomes, this focus neglects many outcomes identified as important by participants in this qualitative study.

Our findings align with Schaffalitzky *et al*'s[24] study with 24 lower limb prosthetic users that revealed that even small gains in mobility provided by a device may constitute a successful outcome from the patient's perspective. Evidence relating to acceptability and use of KAFOs is limited. Swinnen *et al*[2] have reported device functionality and comfort as more highly valued than appearance among 33 patients with neurological conditions (28 of whom used an AFO); views also widely held by participants in our study. Our study findings also echo results from previous studies of patients' views of AFOs[25–27] indicating that orthoses that were ill fitting, considered uncomfortable, heavy, cumbersome and unsightly, and which drew attention to disability were less acceptable to patients.

The important contribution of footwear to people's identity and self-expression is well recognised[28 29] and was a significant concern for both male and female participants in our study, although good fit and comfort were prioritised.

Participants in our study repeatedly emphasised the need for individually tailored care, indicating the need for a service model such as the one advocated by Hutton and Hurry,[30] where orthotic products are not viewed as commodities, but as individually prescribed solutions suited to each patient's personal needs. Lengthy waiting times to access orthotic services and frequent delays in the provision and repair of custom-made KAFOs and footwear were commonly reported in our study, as elsewhere[7 31 32] as impeding device use. Our study participants also emphasised the need for intensive support when receiving a device for the first time, as they adjust to altered self-image. They suggested that in addition to technical information, patients may require psychological support, which has been highlighted previously.[3 30 33]

### Strengths and weaknesses of the study

Our study used robust methods, including the development of the topic guide with a patient and clinicians, and quality assurance of data collection, coding and thematic analysis. The role of the patient advisor in interpretation of the study findings was, however, limited, due to unforeseen circumstances. A full range of neuromuscular and CNS conditions (eg, myasthenia gravis and muscular dystrophy) are not represented in the sample, which is a limitation. Nonetheless, the study included people with regular and sustained contact with orthotic services in connection with knee instability. In addition, we recruited sufficient patients to achieve data saturation,[34] within the limits recommended by Hennink *et al*[35] to achieve *'meaning saturation'*, which enables a richly textured understanding of issues. Inclusion of different age groups revealed some differences in perspectives. Whereas younger participants in our study emphasised the importance of their orthosis in enabling them to undertake full-time employment and function in the role of family breadwinner, older (often retired) people often talked about device use in facilitating their involvement in family and leisure activities that helped them feel connected to the physical and social world beyond their home. We recognise that the views of our study participants may not be reflective of the broader population of people with knee instability due to NMD and CNS conditions; for example, only one person with a diagnosis of stroke was included, and patients younger than 36 years old were absent from the study, and their perspectives warrant investigation. Nor did we set out to systematically assess knee instability in individual patients, using current assessment techniques and grading scales, which may be viewed as a limitation. Rather, we relied on patients to report knee instability as they experienced it, and its implications for daily life. Owing to the comparatively underinvestigated nature of our research topic, transferability of our study findings may be less salient than 'sensitising' readers to

new information,[19] captured through in-depth interviews, promoting new ways of thinking about patients' perspectives of using orthotic devices.

## Applicability to clinical practice and suggestions for further research

The findings have implications for the training of healthcare professionals involved in the prescribing, fitting and follow-up of patients using orthoses to control knee instability who have complex neuromuscular or CNS conditions. To maximise patient acceptance and use of prescribed devices, it is important that the relevant healthcare professionals (orthotists, podiatrists, physiotherapists, occupational therapists) have the appropriate skills, and the necessary time during consultations, to establish a relationship where patients feel listened to, and that supports patients to identify acceptable management strategies and achieve desired treatment outcomes. Patient education and self-efficacy could be enhanced through more flexible routes of referral to orthotic services, including patient self-referral; having a 'named' orthotist as case manager for an individual patient's care, to improve continuity of care; and closely integrated multidisciplinary team working that includes opportunities for shared learning in communications skills.[7 36] More research is needed to assess the impact of such initiatives.

Development and use of a core set of patient-reported outcome measures in the clinical and research setting would facilitate assessment of the effectiveness of different devices and management strategies. Brehm et al[37] have suggested development of a core set of outcomes for studies of lower limb orthoses, based on WHO International Classification of Functioning Disability and Health.[38] To date, however, there has been little qualitative research to 'unpack' patient perspectives of terms such as 'satisfaction' and 'effectiveness', although these constructs have been incorporated into existing outcome measures.[13] We hope that our findings will contribute to the development of outcome measures that reflect patients' priorities.

Our research highlights important aspects that should be included from the patient experience, such as quality of life and psychological well-being, as reflected through ability to engage in a broad range of activities viewed as important for physical and mental well-being, and reduction in pain, trips and falls. It is important that patients are involved in any future work to develop a core outcome set.

## CONCLUSION

Orthotic devices for knee instability play a crucial role in promoting, maintaining and enhancing physical and psychological health and well-being, enabling patients to work, engage in family life and enjoy social activities. Reduction in pain, trips and falls, and improved balance and stability, linked to potential for independent mobility, were regarded as important treatment outcomes. Time constraints and delays in orthotic service delivery can adversely affect timely provision, and patient use, of orthotic devices. Future research should consider how best to measure the impact of orthotic devices on patient quality of life and daily functioning outside the clinic setting, as well as device use and any adverse effects.

**Author affiliations**
[1]York Trials Unit, University of York, York, UK
[2]Department of Health Sciences, University of York, York, UK
[3]Orthotics, Queen Mary's Hospital, St George's University Hospitals NHS Foundation Trust, London, UK
[4]Orthotics/Prosthetics, Royal Children's Hospital Melbourne, Melbourne, Victoria, Australia
[5]Faculty of Allied Health, Midwifery and Social Care, Kingston University/St George's University of London, London, UK
[6]Queen Square Centre for Neuromuscular Diseases, National Hospital for Neurology and Neurosurgery, London, UK
[7]Academic Department of Rehabilitation Medicine, Faculty of Medicine and Health, University of Leeds, Leeds, UK
[8]Rehabilitation, Derby Hospitals/ Nottingham University, Derby, UK
[9]Department of Biomedical Engineering, University of Strathclyde, Glasgow, UK

**Acknowledgements** The authors gratefully acknowledge all the participants who took part in interviews. They would especially like to thank their patient advisor, Sam Bunting, in her role as patient representative. They also wish to acknowledge the contribution of Lisa Dyson to conception of the study.

**Contributors** CM, AB, RB, GR, SL and MP designed the study. RJO provided clinical advice. DM and CJ recruited participants and conducted interviews. Transcripts were coded and analysed by DM with input from CJ. DM wrote the first draft of the manuscript, which was revised with CM and AB. All authors commented on and approved the final manuscript.

**Funding** This project was funded by the National Institute for Health Research (NIHR) HTA Programme (project number 13/30/02) and has been published in full in Health Technol Assess 2016;20(55). Further information available at https://www.journalslibrary.nihr.ac.uk/programmes/hta/133002/#/ This report presents independent research commissioned by the NIHR.

**Disclaimer** The views and opinions expressed by authors in this publication are those of the authors and do not necessarily reflect those of the NHS, the NIHR, MRC, CCF, NETSCC, the HTA programme or the Department of Health.

**Competing interests** During this study, SL was an employee of Opcare, a company that provides orthotic and prosthetic services to the UK NHS. This company does not manufacture orthotic devices, although a sister company ORTHO C FAB does.

**Patient consent for publication** Not required.

**Ethics approval** The study received research ethics (REC reference 14/LO/1132) and governance approvals to recruit patients through NHS sites. The University of York's Department of Health Sciences' Research Governance Committee gave approval to recruit study participants via patient support groups/charities.

**Provenance and peer review** Not commissioned; externally peer reviewed.

**Data availability statement** All data and material related to this research are archived andmaintained according to organisational and ethical regulations. Data are not publiclyavailable due to the risk of participant identification from specific contexts revealedwhen reading entire transcripts and due to the terms and conditions regarding therelease of data to third parties upon which ethical approvals for this study werecontingent. Reasonable requests for further information relating to data can bemade to the corresponding author.

**ORCID iDs**
Alison Booth http://orcid.org/0000-0001-7138-6295

Catriona McDaid http://orcid.org/0000-0002-3751-7260

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
