## [Reviewer comments · BMJ Open]

ARTICLE DETAILS

TITLE (PROVISIONAL)	Orthotic management of instability of the knee related to neuromuscular and central nervous system disorders: a qualitative interview study of patient perspectives
AUTHORS	McCaughan, Dorothy; Booth, Alison; Jackson, Cath; Lalor, Simon; Ramdharry, Gita; O'Connor, Rory J.; Phillips, Margaret; Bowers, Roy; McDaid, Catriona

VERSION 1 – REVIEW

REVIEWER	Sean Sadler University of Newcastle, Australia
REVIEW RETURNED	18-Feb-2019

GENERAL COMMENTS	Thank you for the opportunity to review the manuscript provided. I believe this is an under-explored area and I thank the authors for conducting the study. The manuscript is clearly written and provides valuable insight into patients' perspectives of orthotic therapy for neuromuscular/CNS conditions. I have a few minor comments, outlined below, for the authors to consider. Abstract: -Page 2, line 7: please review the spelling of 'orthoses'-Page 2, line 18: please define 'NHS' on its first use (some readers may be from outside the UK and may not be familiar with this abbreviation).-Please consider adding information to the 'conclusion' section regarding the patient reported difficulties with the orthotic-related services and consider suggesting how future research may address this area (this may not be possible in the abstract, see the final comment under the 'discussion' section below).-Please add that the trial was 'retrospectively' registered, as per the status on the 'ISRCTN registry', Introduction: -Please consider adding more information about how neuromuscular disorders or central nervous systems conditions can result in knee stability, and what is meant by knee instability specifically.-page 3, line 57: After reviewing the "...larger study", reference 10, I wonder if the authors can please provide a brief overview of how the current study is different/what else it offers? Methods: -Please consider adding more detail about the participants and their conditions. For example, the number of people with CMT VS MS, etc.; the duration of the conditions; and the severity/level of impact of these conditions (particularly in regards to their knee instability). I believe this may help provide a more detailed picture of the participants (given the diverse conditions) and may help with the study's generalisability.-Further to the point above, can more information be provided about
--

	the type of orthoses (e.g. leaf spring vs fixed vs hinged), please? Again, I believe this may help with the study's generalisability. -Sampling: besides 'purposive' sampling, can any more information be provided on how specifically participants were selected and how 'instability' was determined? The instability point is something I think could be expanded upon, particularly in context of some of the participants been prescribed AFOs. Is it possible for the authors to provide an overview of how/why the AFOs were prescribed for knee instability (e.g. was it only minor?). I appreciate this may be difficult to determine, given the nature of the study, but would appreciate the authors giving this point some thought, please. -Research ethics: Given that family members were present for some of the interviews (and information they provided was included in some instances), was written consent obtained from these people? I believe the contribution from 'family members' is valuable, but can also see that they may not be directly considered participants of the study (given the intended participants). Please clarify this point. Results: -Please consider the use of the phrase 'the high street'. I wonder if the authors could describe this is a way that that is more universal (given the diverse geographical readership of the journal). Discussion: -I suggest that the authors please consider the following limitations:  -The small number of participants per condition (outside poliomyelitis), and per age group. -The potential for the results to not be able to be applied to a younger cohort (<36 years of age) -Differences in knee instability between participants (if able to be determine as per comment in methods section) or inability to characterise knee instability (if not possible given nature of study). Can this point also be considered in context of the use of AFOs for some participants, please? -Please consider adding 'neuromuscular or central nervous system' to line 33 on page 13, between "complex" and "conditions". Please also consider the general use of "orthotic devices" in this sentence as the findings may not apply to foot related orthotic devices. -Can the authors please consider the need for future research to address service-related issues identified by the participants, and suggest mechanisms by which this may be undertaken? I appreciate the patient-focused approach (in terms of quality of life and daily functioning), but also think that a significant point raised by participants was in regards to the services provided (e.g. fitting/acquisition of their orthosis). I understand this may be beyond the scope of this project, but believe it may be a worthwhile point to identify for future research and think readers may benefit from the authors analysis of this point.
--	---

REVIEWER	Dominique Leibbrandt Stellenbosch University Cape Town South Africa
REVIEW RETURNED	13-Mar-2019

GENERAL COMMENTS	Sampling:  • Where did you find the participants and what was the recruitment process? • More information about the included sample such as gender, age, location, medical history, other treatment received, and functional levels should either be added to the sampling or results section. Transcription: Who transcribed the interviews? Was it one of the researchers or was it an external person? Ethics: The information is given at the end of the manuscript. However, I would suggest adding a line stating that ethical approval was given and that the researchers obtained informed consent from all participants to the methods section of the manuscript. Discussion:  • Might want to consider discussing the role of patient education in addressing concerns and promoting self-efficacy. • Lines 31-35: Please provide some suggestions for how this training can be provided and who should provide it. • Lines 47-49: Why are orthoses important for self-image? Please elaborate Limitations: The studies limitations have not been adequately addressed on page 14. What are the limitations of telephone interviews versus face to face interviews? What are the limitations associated with the purposive sampling approach?
---

REVIEWER	Dr Sarah Drew London School of Economics, UK.
REVIEW RETURNED	23-Apr-2019

GENERAL COMMENTS	Thank you very much for this paper that makes a valuable contribution to a under explored area. My main comments relate to the aims of the study as I don't think that the themes around living with the condition or of accessing services are reflected in the stated aim which is to explore the acceptability and effectiveness of orthoses and treatment outcomes. As a medical sociologist I would have quite liked it if you had brought in some relevant theory but appreciate the value of this descriptive article. In addition, the methodology needs to be discussed in more detail. Abstract The findings section includes a discussion about experiences of living with the condition and accessing services which is not reflected in the abstract. Introduction It would be useful to provide more information about how orthotic devices work, perhaps with diagrams. Numbers of those using devices etc. would also help to place this work in context. It is unclear how many people are affected by these issues. p. 3, l. 35 – 37. I am not sure how relevant it is to cite research that was undertaken in Malawi since this is such a hugely different care
--

	context. Is there any other relevant literature that you could draw on instead such as patient experiences with other medical devices? p. 3, l. 51 – 52. “The range of different outcome measures used to evaluate patient outcomes following device fittings makes it difficult to draw conclusions about effectiveness.” Could you please explain this in more detail? Why does it present a problem? How does this work help to address this? p. 3, l. 57. I would have been interested to have an extra sentence explaining what the wider study is and what it aims to do. Methods p.4, l. 16. How did the patient advisor contribute to the interpretation of study results? More information is needed about how participants were recruited and sampled. How were potential participants identified and approached? How many were approached and declined to participate? This would include those from clinics and those identified through patient support groups. It may be useful to tabulate participant characteristics. p. 4, l. 33. What existing literature informed the topic guide? In the introduction you said that very little work had been conducted on this area. Need a reference for data saturation. p. 4. I think more information is required about what a framework approach, i.e. tabulating data since this is a key feature that differentiates it from other methods. The reader may not be familiar with it. p.5, l. 7. It would be useful to understand how the patient advisor contributed to the interpretation of study findings, perhaps with examples. Results I am not sure that all the themes identified here were inductively generated since they reflect the questions asked in the topic guide that were predefined, e.g. positive and negative aspects of devices. p. 5, l. 11. I am not sure that the impact of living with the conditions is included in the study aims although I appreciate that it is useful to understand these issues to place participants’ goals and outcomes in context. Nevertheless, this needs to be made clear. p. 6. Again, I am not sure that the fitting and acquisition of devices is reflected in the stated study aims as it explores issues of service delivery, e.g. patient-clinician communication. Nevertheless, this is a very interesting and valuable section. Perhaps it would be worthwhile expanding the stated aims of the paper to make it clear that these issues are discussed?
--	--

	p.8, l. 15 – 16. Could you please provide more detail about why patients did not feel the device offered adequate joint support? What does this mean? p.11. The section on desired treatment goals and outcomes has a lot of overlaps with the first theme and some of it feels like repetition. Discussion p.12, Principal findings. These findings do not represent a summary of all the themes identified, only two. There is no summary of findings related to experiences of living with the condition or service delivery which are included in the main findings. Again, I think you need to make it clear that you are aiming to do more in this paper than explore the acceptability, effectiveness and use of devices and outcomes that are important to people. p.12, l. 52. Again, this paragraph explores issues of service delivery. Strengths and weaknesses of the study P. 13, l. 18 – 24. I would have been interested to hear a little more about similarities and differences in experiences between different patient groups, e.g. condition, gender etc. These issues are not discussed in the study findings. Were there differences or were findings broadly the same? Five people were recruited from outside of the NHS clinics through patient support groups. Were these all NHS patients? If they were private, did they have different experiences? I imagine this could impact on experiences of accessing services, e.g. more time in consultations. p.13, l. 26. I am a little bit uncomfortable with the term 'generalisability' as this is quite quantitative. Perhaps 'transferability' is more appropriate. p.13, l. 41 – 53. It would be useful to include a discussion of how these findings compare to the existing outcome measures that were mentioned in the introduction section and how they differ.
--	---

VERSION 1 – AUTHOR RESPONSE

Reviewer 1

Reviewer Name: Sean Sadler

Institution and Country: University of Newcastle, Australia

Competing interests: None declared

Abstract:

-Page 2, line 7: please review the spelling of 'orthoses'

Response: spelling has been amended to 'orthoses'

-Page 2, line 18: please define 'NHS' on its first use (some readers may be from outside the UK and may not be familiar with this abbreviation).

Response: NHS is now defined on its first use, and text now reads 'National Health Service (NHS)'

Please consider adding information to the 'conclusion' section regarding the patient reported difficulties with the orthotic-related services and consider suggesting how future research may address this area (this may not be possible in the abstract, see the final comment under the 'discussion' section below).

Response: We have amended the 'Conclusion' section to include the following sentence (please see Manuscript page 16, lines 17-19):

'Time constraints and delays in delivery of orthotics services can adversely affect timely provision, and patient use, of orthotic devices.'

We have also added a suggestion for further research on patient reported difficulties with orthotics related services, in the section 'Applicability for clinical practice and suggestions for further research.' The Manuscript (page 15, lines 38-43) now reads:

'Patient education and self-efficacy could be enhanced through more flexible routes of referral to orthotic services, including patient self-referral; having a 'named' orthotist as case manager for an individual patient's care, to improve continuity of care; and closely integrated multidisciplinary team working that includes opportunities for shared learning in communications skills (NHS England ref 29 and HEE ref). More research is needed to assess the impact of such initiatives.'

Please add that the trial was 'retrospectively' registered, as per the status on the 'ISRCTN registry'
Response: The text of the Abstract now states that the study 'was retrospectively registered' on the ISRCTN registry.

Introduction:

-Please consider adding more information about how neuromuscular disorders or central nervous systems conditions can result in knee stability, and what is meant by knee instability specifically.

Response: We have added new text into the Introduction in response to these points. The introductory paragraphs with the new text (Manuscript page 3, lines 15-30) now read:

'Adults with knee instability related to neuromuscular disorders (NMD) or central nervous system (CNS) conditions may experience pain, loss of balance, falls and mobility problems due to primary impairments of muscle weakness and/or sensory impairment. Muscle weakness can lead to instability of the joints which can be particularly problematic in lower limbs during weight bearing tasks; if the muscles cannot generate sufficient force to resist gravity, then the lower limb joints can be unstable or collapse. In addition, if sensory feedback is also impaired, people may be unaware of the position of their joints or if they have moved, so increasing the risk of instability.'

Prescription of devices to stabilise the knee will depend on which muscles are weak that act on the knee joint. KAFOs are more often prescribed when proximal lower limb weakness contributes to knee instability, e.g. weakness of the quadriceps. The KAFO will stabilise the position of the femur on the tibia and the tibia over the foot, thus stabilising the knee. If lower forces are required and the ankle is stable, knee braces can be used to stabilise the femur on the tibia to support the knee. In the case of focal distal weakness, e.g. the plantar flexor muscles, AFOs can stabilise the ankle during stance so maintaining the position of the tibia against gravity and preventing a collapse of the knee into excessive flexion or hyper-extension.'

-page 3, line 57: After reviewing the "...larger study", reference 10, I wonder if the authors can please provide a brief overview of how the current study is different/what else it offers?

Response: New text has been inserted into the Manuscript (page 4, lines 27-32), which now reads:

'The report here focuses on treatment goals and outcomes of importance to patients, and device acceptability, contextualised within individuals' experiences of their specific medical condition. Further details of patients' views and experiences of orthotic service delivery can be found in the report of the overall study, which includes results from a systematic literature review, a survey of health-care professionals, and a costing analysis of knee-ankle-foot orthoses.'¹⁰

Methods:

-Please consider adding more detail about the participants and their conditions. For example, the number of people with CMT VS MS, etc.; the duration of the conditions; and the severity/level of

impact of these conditions (particularly in regards to their knee instability). I believe this may help provide a more detailed picture of the participants (given the diverse conditions) and may help with the study's generalisability.

Response: We have now included a new appendix (Appendix 2 Participants' characteristics) which provides this detailed information. Readers are signposted to this information in the Methods section, through addition of the following sentence (page 5, lines 34-36):

'Further details relating to duration and severity of patients' conditions, and reported device use, can be found in Appendix 2: Participant Characteristics'.

-Further to the point above, can more information be provided about the type of orthoses (e.g. leaf spring vs fixed vs hinged), please? Again, I believe this may help with the study's generalisability.

Response: These details relating to devices were not routinely collected. Our exploratory, qualitative study focussed on patients' views and experiences of their orthotic device(s) in relation to mobility and quality of life issues. While some patients spontaneously mentioned 'technical' aspects of their devices, many did not.

-Sampling: besides 'purposive' sampling, can any more information be provided on how specifically participants were selected and how 'instability' was determined?

Response: Details of the study inclusion criteria, and recruitment processes are now included in the Methods section on page 5 of the Manuscript.

'Inclusion criteria for the study were adults (≥ 16 years for NHS participants; > 18 years of age for non-NHS participants) with a neuromuscular disorder who have impaired walking ability primarily due to instability of the knee. Neuromuscular disorder included conditions that primarily affect the peripheral nerve, muscle and neuromuscular junction, for example motor neurone disease, muscular dystrophy, myasthenia gravis, spinal muscular atrophy, CMT disease, poliomyelitis, myopathies and inclusion body myositis. People with knee instability that was related to CNS conditions were also included, for example spinal cord injury, spina bifida and stroke. Participants were people who were able to give informed consent.'

Participants were purposively sampled¹⁶ to reflect a range of conditions, ages, gender, length of time fitted with an orthosis, and regions across England. Potential participants who met the study inclusion criteria were approached by clinicians who were known to them, in three orthotic service/medical rehabilitation NHS outpatient settings, one located in the north, one in the south and one in the middle of England. Patients were informed about the study by the clinicians; those who expressed an interest were given written information about the study, including the researchers' contact details if they wished to learn more and/or discuss taking part in a qualitative interview. Additionally, study information was provided to the chairpersons/lead representatives of the British Polio Fellowship, Charcot-Marie-Tooth UK, the FSH (facioscapulohumeral muscular dystrophy) Support Group UK and the Muscular Dystrophy Campaign, with a request to forward this to their members, who were invited to contact the qualitative researcher directly. Twenty-four people participated, nineteen recruited through NHS clinics and five through charities/patient support groups.

The instability point is something I think could be expanded upon, particularly in context of some of the participants been prescribed AFOs. Is it possible for the authors to provide an overview of how/why the AFOs were prescribed for knee instability (e.g. was it only minor?). I appreciate this may be difficult to determine, given the nature of the study, but would appreciate the authors giving this point some thought, please.

Response: Knee instability in participants ranged along a continuum (mild/moderate/severe) and we have inserted some new text into the Manuscript (page 5, lines 29-30) to indicate this:

'Participants were adults diagnosed with a neuromuscular or CNS condition who had been offered an orthotic device for knee instability of varying degrees of severity, that negatively affected their gait.'

-Research ethics: Given that family members were present for some of the interviews (and information they provided was included in some instances), was written consent obtained from these people? I believe the contribution from 'family members' is valuable, but can also see that they may not be directly considered participants of the study (given the intended participants). Please clarify this point.

Response: Written consent was obtained from all patients who took part in an interview, both those recruited via the NHS and those recruited through charities/patient support groups. Patients who had been interviewed by telephone gave written consent to participation in an interview. We did not deliberately set out to interview relatives though sometimes (on a few occasions) they were present 'by chance' during the patient's interview; where this happened, relatives gave verbal consent that anything they said could be included in any publication or report, as long as it was anonymised.

The section of the Manuscript entitled 'Ethics approval' (page 17, lines 16-20) has now been amended to read:

'Written consent was obtained from all patients who took part in an interview; relatives gave verbal consent for anything they said to be included in any publications. All participants were given assurances concerning confidentiality and anonymity and informed that they could withdraw from the study at any time.'

Results:

-Please consider the use of the phrase 'the high street'. I wonder if the authors could describe this is a way that that is more universal (given the diverse geographical readership of the journal).

Response: The text of the Manuscript (page 12, line 35) has been amended to read:

'Participants using other types of orthoses (such as a knee brace) reported that they usually purchased shoes from local shops selling footwear, preferring particular brands of 'sturdy' shoes.'

Discussion:

-I suggest that the authors please consider the following limitations:

-The small number of participants per condition (outside poliomyelitis), and per age group.

-The potential for the results to not be able to be applied to a younger cohort (<36 years of age)

Response: We recognise these study limitations and have inserted new text into the Manuscript (page 15, lines 17-21) which highlights these limitations:

'We recognise that the views of our study participants may not be reflective of the broader population of people with knee instability due to NMD and CNS conditions; for example, only one person with a diagnosis of stroke was included, and patients younger than 36 years old were absent from the study, and their perspectives warrant investigation.'

-Differences in knee instability between participants (if able to be determine as per comment in methods section) or inability to characterise knee instability (if not possible given nature of study). Can this point also be considered in context of the use of AFOs for some participants, please?

Response: Due to the nature of the study we did not set out to systematically assess knee instability in individual patients, using current assessment techniques and grading scales, which may be viewed as a limitation. Rather, we relied on patients to report knee instability as they experienced it and its implications for daily life. We have therefore inserted the following sentence into the Manuscript (page 15, lines 21-23) for clarification:

'Nor did we set out to systematically assess knee instability in individual patients, using current assessment techniques and grading scales, which may be viewed as a limitation. Rather, we relied on patients to report knee instability as they experienced it, and its implications for daily life.'

-Please consider adding 'neuromuscular or central nervous system' to line 33 on page 13, between "complex" and "conditions". Please also consider the general use of "orthotic devices" in this sentence as the findings may not apply to foot related orthotic devices.

Response: The text of the Manuscript (page 15, lines 32-35) has been amended to read:

'The findings have implications for the training of healthcare professionals involved in the prescribing, fitting and follow-up of patients using orthoses to control knee instability who have complex neuromuscular or central nervous system conditions'

-Can the authors please consider the need for future research to address service-related issues identified by the participants, and suggest mechanisms by which this may be undertaken? I appreciate the patient-focused approach (in terms of quality of life and daily functioning), but also think that a significant point raised by participants was in regards to the services provided (e.g. fitting/acquisition of their orthosis). I understand this may be beyond the scope of this project, but

believe it may be a worthwhile point to identify for future research and think readers may benefit from the authors analysis of this point.

Response: We thank reviewer 2 for highlighting this important aspect of patients' experiences of orthotic service delivery (fitting/acquisition of their orthosis). We did indeed gather a great deal of information relating to patients' perceptions of service provision but due to the limited lengths/word count of the Manuscript, we have not presented this material here in-depth. We have added text into the Manuscript (page 15, lines 38-43) regarding the need for further research into service provision:

'Patient education and self-efficacy could be enhanced through more flexible routes of referral to orthotic services, including patient self-referral; having a 'named' orthotist as case manager for an individual patient's care, to improve continuity of care; and closely integrated multidisciplinary team working that includes opportunities for shared learning in communications skills.^{7, 36} More research is needed to assess the impact of such initiatives.'

Reviewer 2

Reviewer Name: Dominique Leibbrandt

Institution and Country: Stellenbosch University, Cape Town, South Africa

Competing interests: None declared

Please leave your comments for the authors below

Sampling:

- Where did you find the participants and what was the recruitment process?
- More information about the included sample such as gender, age, location, medical history, other treatment received, and functional levels should either be added to the sampling or results section.

Response: We thank reviewer 2 for alerting us to the need to provide more information about sampling in the study. In response to this comment and comments from Reviewer 1 (see above) we have now included information concerning study inclusion criteria and a new appendix (Appendix 2 Participant characteristics) which provides participant details, as also requested by reviewer 2.

Additionally, we have inserted details concerning the recruitment process/study locations (see Manuscript page 5, lines 15-27):

'Potential participants who met the study inclusion criteria were approached by clinicians who were known to them, in three NHS orthotic service/medical rehabilitation outpatient settings, one located in the north, one in the south and one in the middle of England. Patients were informed about the study by the clinicians; those who expressed an interest were given written information about the study, including the researchers' contact details if they wished to learn more and/or discuss taking part in a qualitative interview. Additionally, study information was provided to the chairpersons/ lead representatives of the British Polio Fellowship, Charcot-Marie-Tooth UK, the FSH (facioscapulohumeral muscular dystrophy) Support Group UK and the Muscular Dystrophy Campaign, with a request to forward this to their members, who were invited to contact the qualitative researcher directly.'

Transcription: Who transcribed the interviews? Was it one of the researchers or was it an external person?

Response: Interviews were transcribed by an experienced transcriber working at the University of York who has many years' experience of transcribing qualitative interviews, across a range of studies. The text of the Manuscript has been amended (page 6, lines 13-15) and now reads:

'Audio-recordings were transcribed verbatim by an experienced transcriber, and a random sample of six transcripts checked for accuracy by DM and CJ.'

Ethics: The information is given at the end of the Manuscript. However, I would suggest adding a line stating that ethical approval was given and that the researchers obtained informed consent from all participants to the methods section of the Manuscript.

Response: The following sentence has been inserted into the text of the Manuscript (page 5, lines 38-39):

'Ethics approval was gained for the study to take place and written consent was obtained from all patients who took part in an interview.'

Discussion:

- Might want to consider discussing the role of patient education in addressing concerns and promoting self-efficacy.

- Lines 31-35: Please provide some suggestions for how this training can be provided and who should provide it.

Response: We have inserted some new text into the Manuscript (page 15, lines 38-43) which highlights issues relating to promotion of patient education and self-efficacy:

'Patient education and self-efficacy could be enhanced through more flexible routes of referral to orthotic services, including patient self-referral; having a 'named' orthotist as case manager for an individual patient's care, to improve continuity of care; and closely integrated multidisciplinary team working that includes opportunities for shared learning in communications skills.^{7, 36} More research is needed to assess the impact of such initiatives.'

- Lines 47-49: Why are orthoses important for self-image? Please elaborate

Response: 'Self-image' has been deleted from the Manuscript (Page 14, line 33)

Limitations: The studies limitations have not been adequately addressed on page 14.

(1) What are the limitations of telephone interviews versus face to face interviews?

Response:

The primary concern when comparing telephone and face-to-face interview modes is in the quality of the data collected; that is to say, whether telephone interviews can 'stand-in' for face-to-face interviews without reducing data quality (Sturges and Hanrahan, 2004). Novick (2008) has categorised the types of 'data loss or distortion' that potentially result from the absence of visual cues, the loss of contextual data and the loss or distortion of verbal (spoken) data (Novick, 2008:395). However, Novick calls into question the assumption that data loss or distortion in any of these respects is necessarily detrimental to the interaction or resulting data (Irvine et al., 2013). In our own study, by providing potential participants with a choice between telephone and face-to-face interviewing, we were able to recruit participants whose voices might otherwise not have been heard. Only a small number of interviews were carried out by telephone (3/24) and careful assessment of the data quality between telephone and face-to-face interview revealed no discrepancies, so that we feel assured that that data from telephone interviews were not 'inferior' in any way.

References:

Sturges JE, Hanrahan KJ. Comparing telephone and face-to-face qualitative interviewing: a research note. *Qualitative Research* 2004; 1, 107-118.

Novick G. Is there a bias against telephone interviews in qualitative research? *Research in Nursing and Health* 2008; 31, 391-398.

Irvine A, Drew P, Sainsbury R. 'Am I not answering your questions properly?' Clarification, adequacy and responsiveness in semi-structured telephone and face-to-face interviews. *Qualitative Research* 13,1,87-106.

(2) What are the limitations associated with the purposive sampling approach?

Response:

Two limitations to purposive sampling are noted in the literature: (a) that it is open to researcher bias, and (b) that representativeness/generalisability of findings may be viewed as problematic (Patton, 1990; Polit and Beck, 2010)

(a) Researcher bias: The majority (19) of our study participants were approached about, and recruited into, the study by practising clinicians (orthotists/consultants in rehabilitation medicine) working in NHS clinics; selection of suitable patients was determined by the study inclusion criteria (see below), minimizing potential for researcher bias. A further 5 participants were recruited via charities/patient support groups; information about the study was provided to the chairpersons/lead representatives of the British Polio Fellowship, Charcot-Marie-Tooth UK, the FSH (facioscapulohumeral muscular dystrophy) Support Group UK and the Muscular Dystrophy Campaign, with a request to forward this to their members, who were invited to contact the research team. Those who made contact, and who

were eligible (i.e. met study inclusion criteria, see below), were invited to take part in an interview, thus minimizing potential for researcher bias in the selection of these participants.

Study inclusion criteria: Inclusion criteria for the study were adults (≥ 16 years for NHS participants; > 18 years of age for non-NHS participants) with a neuromuscular disorder who have impaired walking ability primarily due to instability of the knee. Neuromuscular disorder included conditions that primarily affect the peripheral nerve, muscle and neuromuscular junction, for example motor neurone disease, muscular dystrophy, myasthenia gravis, spinal muscular atrophy, CMT disease, poliomyelitis, myopathies and inclusion body myositis. People with knee instability that was related to CNS conditions were also included, for example spinal cord injury, spina bifida and stroke. Participants were people who were able to give informed consent. Principal exclusion criteria were aged < 16 years, people with neuromuscular disorders other than those described above, and people who were unable to give informed consent because of cognitive impairment or for other reasons.

References:

Patton MQ. Qualitative evaluation and research methods. 2nd edition. Newbury Park, CA: Sage, 1990

Polit DF, Beck CT. Generalization in quantitative and qualitative research: Myths and Strategies.

IJNS 2010; 47: 1451-1458.

(b) Representativeness/ generalisability of findings in studies using purposeful sampling

The goal of most qualitative studies is not to generalize, but rather to provide a rich, contextualized understanding of some aspect of human experience through the intensive study of particular cases (Polit and Beck, 2010). Our own small-scale, exploratory study aimed to provide new insights into an area of patient experience (use of an orthotic device for knee instability) hitherto little researched; Green and Thorogood (2018, p308) suggest that 'if researching relatively under-researched topics, or respondents, the issue of generalizability may be less salient than that of 'sensitizing' readers to new ways of thinking or the potential views of respondents'. Our study aimed to provide understanding of factors perceived as significant by the patients using orthotic devices that we spoke to; we do not claim our findings as generalisable to the broader population of device-wearers with NMD and CNS conditions, and we have clarified this in the Manuscript text (page xx, line XX)

Reference:

Polit DF, Beck CT. Generalization in quantitative and qualitative research: Myths and Strategies.

IJNS 2010; 47: 1451-1458.

Green J, Thorogood N. Qualitative Methods for Health Research. 4th ed. London: Sage, 2018, p308.

Response: In line with the Reviewer 2's comments, and to highlight study limitations, we have amended the text of the Manuscript (page 15, lines 2-29) which now reads:

'Our study used robust methods, including development of the topic guide with a patient and clinicians, and quality assurance of data collection, coding and thematic analysis. The role of the patient advisor in interpretation of the study findings was, however, limited, due to unforeseen circumstances. A full range of neuromuscular and CNS conditions (for example, myasthenia gravis and muscular dystrophy) are not represented in the sample, which is a limitation. Nonetheless, the study included people with regular and sustained contact with orthotic services in connection with knee instability. In addition, we recruited sufficient patients to achieve data saturation,³⁴ within the limits recommended by Hennink et al.³⁵ to achieve 'meaning saturation', which enables a richly textured understanding of issues. Inclusion of different age groups revealed some differences in perspectives. Whereas younger participants in our study emphasised the importance of their orthosis in enabling them to undertake full-time employment and function in the role of family breadwinner, older (often retired) people often talked about device use in facilitating their involvement in family and leisure activities that helped them feel connected to the physical and social world beyond their home. We recognise that the views of our study participants may not be reflective of the broader population of people with knee instability due to NMD and CNS conditions; for example, only one person with a diagnosis of stroke was included, and patients younger than 36 years old were absent from the study, and their perspectives warrant investigation. Nor did we set out to systematically assess knee instability in individual patients, using current assessment techniques and grading scales, which may be viewed as a limitation. Rather, we relied on patients to report knee instability as they experienced it, and its implications for daily life. Owing to the comparatively under-investigated nature of our research topic, transferability of our research findings may be less salient than 'sensitising' readers to

new information,¹⁹ captured through in-depth interviews, promoting new ways of thinking about patients' perspectives of using orthotic devices.'

Reviewer 3

Reviewer Name: Dr Sarah Drew

Institution and Country: London School of Economics, UK.

Competing interests: None.

Please leave your comments for the authors below

Thank you very much for this paper that makes a valuable contribution to a under explored area. My main comments relate to the aims of the study as I don't think that the themes around living with the condition or of accessing services are reflected in the stated aim which is to explore the acceptability and effectiveness of orthoses and treatment outcomes. As a medical sociologist I would have quite liked it if you had brought in some relevant theory but appreciate the value of this descriptive article. In addition, the methodology needs to be discussed in more detail.

Abstract

The findings section includes a discussion about experiences of living with the condition and accessing services which is not reflected in the abstract.

Response: We have amended the Abstract to include the following sentences:

'Patients' medical condition impacted significantly on their daily life.'

'Time pressures during device fitting were viewed negatively.'

Introduction

It would be useful to provide more information about how orthotic devices work, perhaps with diagrams.

Response: We have inserted new text into the Manuscript (page 3, lines 25-33) which describes how orthotic devices work. We believe it is beyond the scope of this document to provide diagrams on the biomechanics of each device and the ability to control knee instability.

'Prescription of devices to stabilise the knee will depend on which muscles are weak that act on the knee joint. KAFOs are more often prescribed when proximal lower limb weakness contributes to knee instability, e.g. weakness of the quadriceps. The KAFO will stabilise the position of the femur on the tibia and the tibia over the foot, thus stabilising the knee. If lower forces are required and the ankle is stable, knee braces can be used to stabilise the femur on the tibia to support the knee. In the case of focal distal weakness, e.g. the plantar flexal muscles, AFOs can stabilise the ankle during stance, so maintaining the position of the tibia against gravity and preventing a collapse of the knee into excessive flexion or hyper-extension.'

Numbers of those using devices etc. would also help to place this work in context. It is unclear how many people are affected by these issues.

Response: Data relating to accurate figures in relation to orthotic services use, including device use, is currently lacking. We have inserted some new text into the Introduction section of Manuscript (page 3, 38-40) to this effect, supported by references to relevant literature.

'Accurate figures relating to the number of people treated with orthotic devices are lacking, reflecting the challenges to obtaining data on orthotic services in England, partly due to the complexity of pathways of care^{6,7}

p. 3, l. 35 – 37. I am not sure how relevant it is to cite research that was undertaken in Malawi since this is such a hugely different care context. Is there any other relevant literature that you could draw on instead such as patient experiences with other medical devices?

Response: In line with the reviewer's suggestion, we have removed the reference to the research undertaken in Malawi and introduced details of other recent studies (please see Manuscript page 4, lines 1-19).

'Literature concerning the views and experiences of users of lower limb orthotic devices for knee instability is scant.⁸ A review of 10 studies⁹ examined patient use of lower-limb orthotic devices (e.g. knee brace, AFOs) and orthopaedic shoes, but none of the included studies related to use of KAFOs. The authors reported a wide range of rates of non-use, mainly due to pain, discomfort and non-cosmesis. A recent survey of orthoses users in Belgium² similarly reports varying levels of satisfaction with devices. While inclusion of 'open' questions in the survey elicited 'free text' responses from participants, the underlying reasons for device satisfaction and use were not explored in depth, as this requires the use of appropriate qualitative methods. Results from focus group studies have appeared recently, but do not elucidate KAFO-users' perspectives. Swinnen et al.¹⁰ conducted 4 focus groups with 20 patients diagnosed with multiple sclerosis that revealed that stigmatization, difficulties in putting on their lower limb orthosis (AFO) and aesthetic aspects, were implicated in non-use. A focus group with eight AFO users living in the Netherlands to gain insight into importance of device related activities to AFO-users found that participants ranked walking as the most important of 11 different activities.¹¹ However, this latter small study did not include any KAFO users, whose priorities for treatment outcomes may differ.'

p. 3, l. 51 – 52.

"The range of different outcome measures used to evaluate patient outcomes following device fittings makes it difficult to draw conclusions about effectiveness."

Could you please explain this in more detail? Why does it present a problem? How does this work help to address this?

Response: Outcome measures are useful in assessment, clinical decision making and evidencing the outcomes of treatment to either the service user or third parties. They also facilitate clinical audit and research (please see <https://www.bapo.com/wp-content/uploads/2019/02/Measuring-Change-BAPO-website.pdf>). The lack of consistent outcome measures (and their inconsistent application in neurological patients) makes it difficult to evaluate different orthoses and draw comparisons. We would hope that our work would contribute to the development of patient-focussed outcome measures that include outcomes of importance to patients. We have amended the Manuscript text (page 4, lines 23-24) to read:

'Understanding of the treatment outcome priorities of people with knee instability related to neuromuscular and CNS conditions is limited, and the range of different outcome measures used to evaluate patient outcomes following device fittings^{12,13} makes it difficult to draw conclusions about effectiveness as data cannot be pooled for meta-analysis.'

p. 3, l. 57. I would have been interested to have an extra sentence explaining what the wider study is and what it aims to do.

Response: The sentence below has inserted into the text of the Manuscript (page 4, lines 27-33) 'The report here focuses on treatment goals and outcomes of importance to patients, and device acceptability, contextualised within individuals' experiences of their specific medical condition. Further details of patients' views and experiences of orthotic service delivery can be found in the report of the overall study, which includes results from a systematic literature review, a survey of health-care professionals, and a costing analysis of knee-ankle-foot orthoses.'⁸

Methods

p.4, l. 16. How did the patient advisor contribute to the interpretation of study results?

Response: We have deleted the sentence referring to contribution of the patient advisor to interpretation of the study results as their role was compromised due to unforeseen circumstances (i.e. illness). We have highlighted this as a limitation of the study in the Manuscript (page 15, lines 3-4).

'The role of the patient advisor in the interpretation of the study findings was, however limited, due to unforeseen circumstances.'

More information is needed about how participants were recruited and sampled. How were potential participants identified and approached? How many were approached and declined to participate? This would include those from clinics and those identified through patient support groups. It may be useful to tabulate participant characteristics.

Response: We have inserted details of recruitment processes into the Manuscript (please see Manuscript page 5, lines 15-26).

'Potential participants who met the study inclusion criteria were approached by clinicians who were known to them, in three NHS orthotic service/medical rehabilitation outpatient settings, one located in the north, one in the south and one in the middle of England. Patients were informed about the study by the clinicians; those who expressed an interest were given written information about the study, including the researchers' contact details if they wished to learn more and/or discuss taking part in a qualitative interview. Additionally, study information was provided to the chairpersons/ lead representatives of the British Polio Fellowship, Charcot–Marie–Tooth UK, the FSH (facioscapulohumeral muscular dystrophy) Support Group UK and the Muscular Dystrophy Campaign, with a request to forward this to their members, who were invited to contact the qualitative researcher directly.'

Response: Additionally, we have included a new appendix (Appendix 2) which gives details of participants' characteristics.

p. 4, l. 33. What existing literature informed the topic guide? In the introduction you said that very little work had been conducted on this area.

Response: The section relating to development of the topic guide has been revised and a new reference inserted. The Manuscript (page 5, line 42-page 6, lines 1-5) now reads:

'A topic guide was developed, based on the aims of the research to explore patient perceptions of using an orthosis and goals for treatment; we sought to develop an instrument sufficiently structured to ensure consistency in information gathering, but flexible enough to allow participants to recount their individual experiences.¹⁷ Input from the patient advisor helped guide relevance of content and comprehensiveness of the topic guide.'

Need a reference for data saturation.

Response: A reference is provided for data saturation (reference 34), which is included in the list of references: Saunders B, Sim J, Kingstone T, et al. Saturation in qualitative research: exploring its conceptualization and operationalization. *Qual Quant* 2018; 52:1893–1907

p. 4. I think more information is required about what a framework approach, i.e. tabulating data since this is a key feature that differentiates it from other methods. The reader may not be familiar with it.

Response: The following text has been inserted into the Manuscript to clarify the charting or tabulation stage of Framework approach (see Manuscript page 6, lines 19-22). The new text is supported with a reference (reference 19)

'Framework approach incorporates 'charting' or tabulation of data, which involves summarising and re-arranging data according to thematic content, enabling the analyst to easily see the range of data across cases and under themes.¹⁹

p.5, l. 7. It would be useful to understand how the patient advisor contributed to the interpretation of study findings, perhaps with examples.

Response: Please also see response to Reviewer 3's comment relating to page 4, line 16 above. The Manuscript has now been amended to read (page 15, lines 3-4)

'The role of the patient advisor in interpretation of the study findings was, however, limited, due to unforeseen circumstances.'

Results

I am not sure that all the themes identified here were inductively generated since they reflect the questions asked in the topic guide that were predefined, e.g. positive and negative aspects of devices.

Response: As is often the case in qualitative research, our analytic strategy was neither purely inductive or deductive, but a balance of both orientations (Green and Thorogood, 2018, page 252). Harding (2018, page 34) refers to 'an inductive-deductive continuum', along which researchers may move back and forth during the process of data analysis. We sought to derive themes and explanations (inductive approach) from a close reading of our data derived from the patient interviews, but were guided in this search by the questions we were seeking to answer in the study (deductive approach) that were manifest in the topic guide used in the interviews. We have inserted new text into the Manuscript on page 6, lines 25-27) to reflect this.

'Guided by the research questions, our approach to data analysis balanced both inductive and deductive orientations, as we sought to transfer the 'raw' data into a new and coherent description of the phenomena under scrutiny.²¹

References:

Green J and Thorogood N. Qualitative Methods for Health Research. 4th ed. London: Sage, 2018, page 252.

Harding J. Qualitative Data Analysis. 2nd ed. London: Sage Publications, 2018, page 34.

p. 5, l. 11. I am not sure that the impact of living with the conditions is included in the study aims although I appreciate that it is useful to understand these issues to place participants' goals and outcomes in context. Nevertheless, this needs to be made clear.

Response: We have amended the Manuscript (page 4, lines 27-33) in line with this comment to highlight that participants' views on treatment goals and outcomes are framed by their experiences of their specific medical condition.

'The report here focuses on treatment goals and outcomes of importance to patients, and device acceptability, contextualised within individuals' experiences of their specific medical condition. Further details of patients' views and experiences of orthotic service delivery can be found in the report of the overall study, which includes results from a systematic literature review, a survey of health-care professionals, and a costing analysis of knee-ankle-foot orthoses.⁸

p. 6. Again, I am not sure that the fitting and acquisition of devices is reflected in the stated study aims as it explores issues of service delivery, e.g. patient-clinician communication. Nevertheless, this is a very interesting and valuable section. Perhaps it would be worthwhile expanding the stated aims of the paper to make it clear that these issues are discussed?

Response: We have amended the text in the Abstract and the Manuscript regarding the study aims to reflect that we have reported information relating to patients' views of fitting/acquisition of their orthotic device.

The Abstract now includes the following sentences:

'Our study aimed to elicit device users' perspectives regarding fitting, acceptability, effectiveness and use of orthoses, and identify important treatment outcomes.'

'Time pressures during device fitting were viewed negatively.'

The study aims in the Manuscript (page 4, lines 24-27) have been amended to read:

'Our study aimed to: (i) explore patient perceptions regarding fitting/acquisition, acceptability, effectiveness and usage of orthoses, and (ii) identify the outcomes that are important for people who have been fitted with an orthotic device for knee instability, across a broad range of conditions.'

p.8, l. 15 – 16. Could you please provide more detail about why patients did not feel the device offered adequate joint support? What does this mean?

Response: We have amended the Manuscript (page 9, line 36) to read:

'Three of the 24 participants reported almost no use of their device as it did not offer adequate support for their knee or ankle.'

p.11. The section on desired treatment goals and outcomes has a lot of overlaps with the first theme and some of it feels like repetition.

Response: Impact of their condition on patients and their desired treatment goals were inextricably linked. We take account of the Reviewer's comment however and have removed some text/quotations from section 5.Desired treatment goals and outcomes (Manuscript page 13, lines 15-18). The following sentences have been removed from the text:

'Walking is essential to him...to keep active as much as possible' (Wife of P9)

Many people emphasised being able to continue to drive their care as vital for work and leisure.

'It's very important having a car, I couldn't do my job without a car (P3)

'I want to drive down and pick mum up and bring her here' (P10)

Discussion

p.12, Principal findings. These findings do not represent a summary of all the themes identified, only two. There is no summary of findings related to experiences of living with the condition or service delivery which are included in the main findings. Again, I think you need to make it clear that you are aiming to do more in this paper than explore the acceptability, effectiveness and use of devices and outcomes that are important to people.

p.12, l. 52. Again, this paragraph explores issues of service delivery.

Responses to above two comments: We have expanded the section describing the Principal Findings through the insertion of new text referring to patient experiences of living with the condition and patient perceptions of service delivery (please see Manuscript page 13, lines 34-35, and page 14, lines 7-15)

'Participants described knee instability as compromising their ability to pursue desired daily activities and as having a negative impact on their social life.'

'Participants highlighted various issues relating to the fitting/acquisition of devices, such as long waits between appointments, and time pressures during appointments, that resulted in reduced opportunities for contact and communication with clinicians. As a result, the period over which fitting took place might be prolonged, and patients were sometimes left feeling they did not receive the support and education they required to optimise adaptation to their device, especially where the device was a KAFO.'

Strengths and weaknesses of the study

P. 13, l. 18 – 24. I would have been interested to hear a little more about similarities and differences in experiences between different patient groups, e.g. condition, gender etc. These issues are not discussed in the study findings. Were there differences or were findings broadly the same?

Response: They were broadly the same; we noted that men in our study were equally concerned as women with device and shoe appearance. We have amended the manuscript (page 14, lines 7-8) to include the sentence:

'Male participants were as equally concerned as females with cosmetics of devices and shoes.'

Five people were recruited from outside of the NHS clinics through patient support groups. Were these all NHS patients? If they were private, did they have different experiences? I imagine this could impact on experiences of accessing services, e.g. more time in consultations.

Response: Although these five participants were recruited via support groups rather than through NHS clinics, they nonetheless had accessed/been treated through NHS provided orthotic services.

The new text which has been added to the Manuscript (page 5, lines 15-26) describing recruitment processes in more detail should help to clarify this point.

p.13, l. 26. I am a little bit uncomfortable with the term 'generalisability' as this is quite quantitative. Perhaps 'transferability' is more appropriate.

Response:

We recognise that the term 'transferability' is more frequently used in reporting qualitative research; we are also aware that the limitations of our study call for caution in relation to extrapolation of findings. We have now inserted the following sentence into the Manuscript (page 15, lines 17-21): 'We recognise that the views of our study participants may not be reflective of the broader population of people with knee instability due to NMD and CNS conditions; for example, patients younger than 36 years old were absent from the study, and their perspectives warrant investigation'

Additionally, as suggested, we have substituted the word 'transferability' for 'generalisability' in the following sentence (page15, line 26):

'Owing to the comparatively under-investigated nature of our research topic, transferability of our study findings may be less salient than 'sensitising' readers to new information,¹⁶ captured through in-depth interviews, promoting new ways of thinking about patients' perspectives of using orthotic devices.'

p.13, l. 41 – 53. It would be useful to include a discussion of how these findings compare to the existing outcome measures that were mentioned in the introduction section and how they differ.

Response: In line with the reviewer's suggestion, we have amended the Discussion section of the Manuscript (page 16, lines 3-7) to incorporate reference to our findings in relation to other outcome measures mentioned in the Introduction. The Manuscript now reads:

'To date however, there has been little qualitative research to 'unpack' patient perspectives of terms such as 'satisfaction' and 'effectiveness', although these constructs have been incorporated into existing outcome measures.¹³ We hope that our study findings will contribute to future development of outcome measures that reflect patients' priorities.'

VERSION 2 – REVIEW

REVIEWER	Sean Sadler University of Newcastle, Australia
REVIEW RETURNED	23-Aug-2019
GENERAL COMMENTS	Thank you for taking the time to thoroughly and clearly address my comments/suggestions. By providing the example of what was edited/added made the reviewing process much easier, thank you.
REVIEWER	Dominique Leibbrandt Stellenbosch University
REVIEW RETURNED	28-Aug-2019
GENERAL COMMENTS	All of my concerns have been adequately addressed by the authors